# Cross-National variations in mental health: A cross-sectional study on depression, anxiety, and stress among university staff and students in Sub-Saharan African

Oforbuike Onyebuchi Ike[1], Michael Agyemang Kwarteng[2], Grace Ogbonna[3], Isaura Brito dos Santos[4]°, Osamudiamen McHillary Ogiemudia[5], Anayochukwu E. Anyasodor[6], Ellen Konadu Antwi-Adjei[7], Okechi Ulumma Amaechi[8], Ejididke Gertrude Ebele[9], Ngozika Esther Ezinne[10,11], Khathutshelo Percy Mashige[12], Antor Ndep[13], Edith Daniel-Nwosu[14]°, Bernadine Nsa Ekpenyong[15], Tshubelela Sello Simon Magakwe[16], Kingsley E. Agho[1718], Ayomikun Nifemi Dahunsi[19], Oyelola A. Adegboye[20], Kelechi C Ogbuehi[21], Uchechukwu Levi Osuagwu[18,22]*, Center for eyecare and Public Health Intervention Initiative (CEPHII)[11]

**1** Department of Optometry, Bayero University Kano, Nigeria, **2** Optometry Unit, Department of Clinical Surgical Sciences, Faculty of Medical Sciences, The University of the West Indies, St Augustine Campus, Trinidad and Tobago, **3** Department of Optometry, Mzuzu University Malawi, Luwinga, Mzuzu, **4** Department of Community Health, Faculty of Medicine, Eduardo Mondlane University, Maputo, Mozambique, **5** Department of Optometry, Faculty of Life Sciences, University of Benin, and Department of Optometry, College of Medical and Health Sciences, Novena University, Kwale, Benin City, Delta State, Nigeria, **6** Rural Health Research Institute, Charles Sturt University, Orange, New South Wales, Australia, **7** Department of Optometry and Visual Science, College of Science, Kwame Nkrumah University of Science and Technology, Kumasi, Ghana, **8** Department of Optometry, Faculty of Health Sciences, Abia State University, Uturu, Okigwe, Abia State, Nigeria, **9** Department of Optometry, Madonna University, Elele, Rivers State, Nigeria, **10** Department of Clinical Surgical Sciences, Faculty of Medical Sciences, Optometry Unit, The University of the West Indies, St Augustine Campus, Trinidad and Tobago, **11** Bathurst Rural Clinical School, Western Sydney University, Bathurst New South Wales, Australia, **12** Department of Optometry, African Vision Research Institute, University of KwaZulu-Natal, Durban, South Africa, **13** Health Education & Health Promotion Unit, Department of Public Health, Faculty of Allied Medical Sciences, College of Medical Sciences, University of Calabar, Calabar, Cross River State, Nigeria, **14** Department of Optometry, Federal University of Technology Owerri, Owerri, Imo State, Nigeria, **15** Department of Public Health, Faculty of Allied Medical Sciences, College of Medical Sciences, University of Calabar, Calabar, Cross River State, Nigeria, **16** Discipline of Optometry, University of KwaZulu-Natal, Durban, South Africa, **17** School of Health Science, Western Sydney University, Campbelltown, New South Wales, Australia, **18** Discipline of Optometry, African Vision Research Institute,University of KwaZulu-Natal, Durban, South Africa, **19** Millenium Eye Center, University of Medical Sciences Teaching Hospital, Akure, Ondo State, Nigeria, **20** Menzies School of Health Research, Charles Darwin University, Northern Territory, Australia, **21** Department of Medicine, Dunedin School of Medicine, University of Otago, Dunedin, New Zealand, **22** Bathurst Rural Clinical School (BRCS), School of Medicine, Western Sydney University, Bathurst, New South Wales, Australia

☯ These authors contributed equally to this work.
\* l.osuagwu@westernsydney.edu.au

## Abstract

### Purpose

Mental health disorders are global concerns, but their impact varies across regions. In sub-Saharan Africa (SSA), the influence of country-specific factors on mental

**Data availability statement:** All relevant data are within the paper and its Supporting Information files.

**Funding:** The author(s) received no specific funding for this work.

**Competing interests:** The authors have declared that no competing interests exist.

health is under-researched. This study investigates the influence of country of origin on the prevalence and severity of mental health conditions among university students and staff across select SSA countries.

## Method

A cross-sectional, web-based survey using the Depression, Anxiety, and Stress Scale-21 (DASS-21) was conducted from 16 April to 18 November 2024. The survey was distributed online through multiple African social networks, reaching students and staff from different universities across Africa. Prevalence estimates for anxiety, depression, and stress were based on binomial distribution with Clopper-Pearson confidence intervals, while country-level differences were assessed using univariate odds ratios and multiple logistic regression.

## Results

Of the 3221 participants, aged 25.3±8.6 (mean±SD), the majority (1850, 57.3%) were females. Findings revealed that Nigerians reported the highest prevalence of severe and extremely severe mental health conditions, while Ghanaians recorded the lowest levels across all mental health categories. Multivariable analysis revealed that, compared to Ghanaians, respondents from Malawi had the strongest odds for mental health challenges followed by Mozambique and Nigeria. The adjusted odds ratios (AORs) for Malawi were 4.39 (95% CI: 3.28–5.89), 3.86 (95% CI: 2.81–5.29) and 4.51 (95% CI: 3.33–6.11) for depression, anxiety, and stress, respectively.

## Conclusions

This study found significant differences in mental health outcomes between participants from Ghana, Malawi, Mozambique, and Nigeria. Malawi had the greatest risk for depression, anxiety, stress, and combinations of those conditions while Ghana reported the lowest risks. The findings emphasize the importance of considering contextual factors, such as education levels, gender, and country of origin, in understanding mental health risks. Overall, these findings underscore the critical mental health burden in sub-Saharan Africa and the need for increased access to mental health resources and targeted interventions.

## Introduction

Mental health disorders, including depression, anxiety and stress, are a growing public health concern, with significant global prevalence [1,2] which varies across populations [3,4]. For example, the global incidence of depression increased from 172 million cases in 1990 to 258 million in 2017, marking a 49.86% rise [5,6]. According to the World Health Organization [7], approximately 280 million people worldwide suffer from depression, a condition that disproportionately affects women and is often linked

to economic burdens, stigma, discrimination, and social exclusion [8,9]. Mental health disorders affect individuals across all ages, genders, and socioeconomic backgrounds, exerting a widespread impact across diverse geographic and demographic groups [10]. In sub-Saharan African (SSA) countries, the burden of depression, stress, and anxiety in the university sub-population is high [11–13] highlighting the need to address this group to establish solutions for their wellbeing.

Students and staff in universities within under-resourced educational systems in low- and middle-income countries are especially susceptible to mental health challenges, as they contend with distinctive stressors such as heavy academic workloads and intense pressure to perform [14–16]. Researching these groups offers valuable insights into their mental health challenges, enabling targeted interventions to improve health, academic performance, and workplace productivity, with benefits extending to society at large.

However, the current body of literature, frequently neglects the role of cross-cultural differences among the university sub-population across SSA countries [17,18], leading to a significant gap in understanding. By examining the influence of country of origin on the prevalence and severity of mental health conditions such as depression, anxiety, and stress among university staff and students across SSA countries, this study aims to bridge the existing knowledge gap. Specifically, the objectives of this study include determining the prevalence of depression, anxiety, and stress among university staff and students across different sub-Saharan African countries, identifying significant cross-national variations, and assessing the role of demographic variables in influencing mental health outcomes across these SSA countries.

## Materials and methods

### Participants

The participants in this study included students and staff affiliated with various universities and colleges across four SSA countries [Nigeria, Ghana, Malawi, and Mozambique]. Individuals associated with African institutions were targeted for recruitment.

### Settings

Nigeria, with a GDP of $252.7 billion in 2022 and a per capita GDP of $1,100, has Africa's largest population [19]. Ghana, with a population of about 34.8 million, had a GDP of $73.36 billion in 2021, making it the eighth-largest economy in Africa. The GDP per capita was estimated at nearly $2,300, though population growth continues to influence income growth per capita [19]. Malawi's population is characterized by a young age structure, a high dependency ratio, and a nearly equal sex ratio. Life expectancy remains relatively low, and extreme poverty affects 71% of the population. The economy relies heavily on rainfed and primary agriculture, limiting inclusive growth [20]. Mozambique, with a population of 33.0 million in 2022, had a GDP of $17.9 billion and a per capita GDP of $543.5. Poverty remains widespread, with 64.6% of the population living below the international poverty line ($2.15/day) and 83.1% below the lower middle-income threshold ($3.65/day) [21].

### Study design and procedure

This was a web-based, cross-sectional survey that utilized a convenience sampling method for data collection which took place from Tuesday, April 16th to Monday, November 18, 2024, using a validated questionnaire. Before participation, all potential participants were provided with detailed information about the study, including its nature and purpose, through an online preamble. The questionnaire was translated and administered in Portuguese and English languages via Google Forms, across SSA countries. To ensure efficient distribution and accessibility, an electronic link to the survey was shared via social media platforms, including WhatsApp and Facebook, as well as through the authors' email contacts. The survey distribution primarily utilized a snowballing approach, leveraging virtual networks to reach individuals who engaged with social media and other online platforms. This method was both time- and cost-effective for data collection [22,23]. The use

of an online survey facilitated a broad outreach to a large cohort across SSA, enabling timely data collection with limited resources. Moreover, the use of social media platforms significantly allowed for the rapid dissemination of survey information [24]. The reporting of this study strictly adhered to the STrengthening the Reporting of OBservational studies in Epidemiology (STROBE) guidelines for cross-sectional studies [25] (Supplementary File 1)

## Questionnaires

A validated, self-administered survey (Supplementary File S2) was adapted to align with the objectives of this study. The first section of the survey collected data on the participants' sociodemographic characteristics, including age, gender, country of origin, institutional affiliation, location, year of study, department, and name of school. Students were asked to indicate their course of study, while staff reported the school in which they were employed. Additional variables included the highest level of educational qualification, marital status, and faculty designation (i.e., student, academic, or non-academic staff).

## DASS-21 Survey scale

This is a short version of the Depression, Anxiety, and Stress Scale-21 (DASS-21), utilized for psychological screening purposes [26]. The instrument, which has been previously validated [27–29] and pretested in this study, is specifically designed to distinguish between symptoms of depression, anxiety, and stress. Each of the three DASS-21 scales consists of 7 items, grouped into subscales with similar content. Responses were scored using a 4-point Likert scale ranging from 0 to 3, where 0 indicates "did not apply to me at all", 1 indicates "applied to me to some degree", 2 indicates "applied to me to a considerable degree", and 3 indicates "applied to me very much". In the assessment of depression, a score of 0–9 on the depression subscale was classified as normal, 10–13 as mild, 14–20 as moderate, 21–27 as severe, and scores of 28 or higher as extremely severe. For anxiety, a score of 0–7 was considered normal, 8–9 as mild, 10–14 as moderate, 15–19 as severe, and 20 or above as extremely severe. Regarding stress, scores of 0–14 were classified as normal, 15–18 as mild, 19–25 as moderate, 26–33 as severe, and scores 34 and above as extremely severe.

The questionnaire was pre-tested one week before the main data collection on a sample of 30 participants, who were not included in the final survey. Following the pre-test, the questionnaire was revised based on the feedback received. The Cronbach's alpha (α) coefficient for the DASS-21 scale was determined to be 0.82, indicating good internal consistency across all items.

## Inclusion and exclusion criteria

Survey responses from individuals who were not originally from SSA but were currently studying or working in the region were excluded from the analysis. Additionally, responses originating from identical Internet Protocol addresses and exhibiting similar demographic characteristics, such as age and gender, were considered duplicate entries. In such cases, only the most complete entry was retained for analysis. Participants who provided informed consent were included.

## Sample size determination

The desired sample size was calculated [30] considering a precision level of 3.5%, a non-response rate of 10%, a 95% confidence level, and a proportion of adults with mental health as 50% since a similar cross-national study has not been conducted.

This yielded a sample size of 3484 participants. This sample size was considered sufficient to detect statistically significant differences in the analysis of an online cross-sectional study investigating mental health symptoms among university staff and students in SSA. However, a total of 3227 (92.6%) responses were included in the analysis.

### Reliability test

The reliability of the survey tools was confirmed with sufficient Cronbach's alpha coefficients for the DASS-21. The DASS-21 scale showed a reliability coefficient of 0.942, with a total of 21 items. The subscales for depression, anxiety, and stress had reliability coefficients of 0.806, 0.873, and 0.871, respectively, with each subscale containing 7 items.

### Data analysis

The study utilized the DASS-21 survey to measure three outcome variables—depression, anxiety, and stress—which were analysed as categorical variables based on classification thresholds. Mental health conditions were further dichotomized into binary categories: 0–9 ("No depression") and 10+ ("Major depression") for depression; 0–7 ("No anxiety") and 8+ ("Anxiety") for anxiety, and 0–14 ("No stress") and 15+ ("Stress") for stress. Predictor variables included country of origin, focusing on Ghana, Nigeria, Malawi, and Mozambique due to sufficient sample sizes. Covariates encompassed sociodemographic factors such as age, gender, marital status, education level, faculty category (student, academic staff, non-academic staff), religious affiliation, level of study, rurality, and country of residence.

The sample characteristics were presented using frequencies and percentages. Data from other participating countries were excluded from the analysis due to the limited number of respondents. Only countries with a minimum of 300 respondents were included in the analysis. The prevalence of anxiety, depression, and stress were estimated using the binomial distribution with Clopper-Pearson confidence intervals. To assess country-level differences in risk of anxiety, depression, stress, and their combination, odds ratios (ORs) were calculated using bivariate analysis to examine associations with the country of residence. Multivariable logistic regression was subsequently performed to investigate these associations further. The multivariable analysis was adjusted for the following variables: age group, gender, marital status, educational level, ethnicity, religion, location of workplace, and having children. Statistical significance was assessed at a 5% significance level. All statistical analyses were performed using Statistical Package for the Social Sciences (IBM Corp, SPSS Statistics for Windows Version 21.0, Armonk, NY, USA) and R version 4.0.3.

### Ethical consideration

Ethical approval was obtained from several institutions, including the Research Ethics Committee of the Federal University of Technology Owerri, Nigeria (FUT/SOHT/REC/vol. 4/2), the Research Ethics Committee of Abia State University Uturu, Nigeria (ABSU/REC/OPT/002/2024), the Health Research Ethics Committee of Bayero University Kano, Nigeria (NHREC/BUK-HREC/476/10/2311) and the Committee on Human Research, Publication and Ethics, Kwame Nkrumah University of Science and Technology, Ghana (CHRPE/AP/374/24). Written informed consent was obtained electronically from all participants by asking them to respond with a "Yes" or "No" to the question, "Do you consent to voluntarily participate in this survey?" Consent was implied when participants selected the "Yes" option. To ensure this, the consent question was made mandatory, preventing participants from advancing in the questionnaire without responding.

## Results

### Characteristics of the study population

The flowchart in Fig 1: "Flowchart of Respondent Selection Process in Study Analysis" illustrates the data process steps used in this study. After removing duplicates, non-African respondents, and countries with insufficient and incomplete data, a final sample of 3,221 participants was obtained, with the majority from Nigeria (1832, 56.9%) and Ghana (596, 18.5%). The mean±SD age of the participants was 25.3±8.6 years (range, 15–72 years) (Table 1). The majority of respondents were students (n = 2807, 87.0%) and females (1845, 57.3%).

Most respondents identified as Christian (2691, 83.4%) and not married (2741, 85.1%), with nearly all respondents (96.4%) reporting African descent. Regarding educational attainment, 63.9% (n = 2059) held Diploma and lower

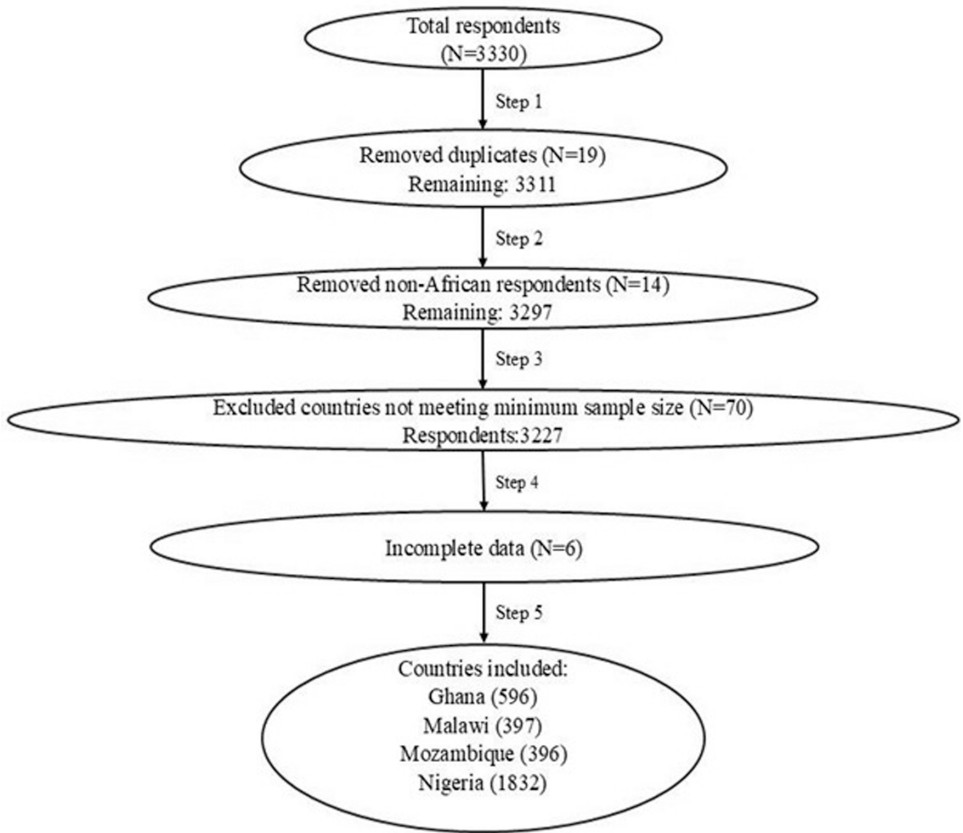

**Fig 1. Flowchart of respondent selection process in study analysis.**

qualifications, while 36.1% (n = 1162) had obtained a bachelor's degree and above. Additionally, 83.7% (n = 2639) of respondents indicated that they did not have children. In terms of work location, 38.3% (n = 1237) were employed in urban areas, although 36.8% of respondents had their residence unclassified.

**Prevalence and distribution of severity of mental health conditions.**

Mental health condition prevalence varied significantly across Ghana, Malawi, Mozambique, and Nigeria (Table 2). Malawi had the highest rates of depression (72.3%, 95 CI: 67.6–76.6), anxiety (81.4%, 95%CI: 77.2–85.1), stress (50.6%, 95%CI: 45.6–55.7), and co-occurring conditions, while Ghana had the lowest across most categories. The overall prevalence across all four countries was 53.3% (95%CI: 51.5–55.0) for depression, 63.5% (95%CI: 61.8–65.2) for anxiety, and 30.1% (95%CI: 28.6–31.8) for stress. Co-occurring conditions were most common in Malawi (72.3%, 95%CI: 67.6–76.6) and least prevalent in Ghana (38.9%, 95%CI: 34.9–42.9). The prevalence of individuals experiencing all three conditions was highest in Malawi (45.3%, 95%CI: 40.4–50.4) and lowest in Ghana (17.3%, 95%CI: 14.3–20.6), with Mozambique and Nigeria showing elevated rates in specific categories.

Fig 2 illustrates the distribution of respondents by their depression, anxiety, and stress levels, among staff and students of African universities. It shows that a significant proportion of respondents experienced moderate to severe levels of mental health disorders, particularly anxiety (Fig 2 top panel). Anxiety was predominantly categorized as 'extremely severe' in 27.6% of respondents (n = 893), with a lower percentage falling into the 'normal' anxiety category (n = 1175, 36.5%).

**Table 1. Distribution of socio-demographics by country of residence.**

| Variables | Sub-group | Country of Residence, n (%) | | | | N (%) | P-value |
|---|---|---|---|---|---|---|---|
| | | Ghana | Malawi | Mozambique | Nigeria | | |
| Age group (years) | Younger age (<18yrs) | 10 (0.3) | 1 (0.0) | 7 (0.2) | 41 (1.3) | 59 (1.8) | <0.001 |
| | Youth (18–35yrs) | 547 (17.0) | 368 (11.4) | 322 (10.0) | 1552 (48.2) | 2789 (86.6) | |
| | Adult and Elderly (≥36yrs) | 39 (1.2) | 28 (0.9) | 67 (2.1) | 239 (7.6) | 373 (11.8) | |
| Gender, n=3197 | Female | 353 (11.0) | 177 (5.5) | 241 (7.5) | 1,074 (33.6) | 1,845 (57.7) | <0.001 |
| | Male | 236 (7.4) | 218 (6.8) | 151 (4.7) | 747 (23.4) | 1,352 (42.3) | |
| Ethnicity | African | 583 (18.1) | 395 (12.3) | 347 (10.8) | 1779 (55.2) | 3104 (96.4) | <0.001 |
| | Others | 13 (0.4) | 2 (0.1) | 49 (1.5) | 53 (1.6) | 117 (3.6) | |
| Job description, n=3192 | Academic staff | 40 (1.3) | 16 (0.5) | 49 (1.5) | 198 (6.2) | 303 (9.5) | <0.001 |
| | Non-academic staff | 23 (0.7) | 7 (0.2) | 11 (0.3) | 47 (1.5) | 88 (2.8) | |
| | Student | 530 (16.6) | 373 (11.7) | 336 (10.5) | 1568 (49.1) | 2807 (87.9) | |
| Marital status | Married | 35 (1.1) | 51 (1.6) | 75 (2.3) | 319 (9.9) | 480 (14.9) | <0.001 |
| | Not married | 561 (17.4) | 346 (10.7) | 321 (10.0) | 1513 (47.0) | 2741 (85.1) | |
| Religion | Christianity | 526 (16.3) | 380 (11.8) | 324 (10.1) | 1455 (45.2) | 2691 (83.4) | <0.001 |
| | Islam | 57 (1.8) | 11 (0.3) | 56 (1.7) | 347 (10.8) | 471 (14.6) | |
| | Others | 13 (0.4) | 6 (0.2) | 16 (0.5) | 30 (0.9) | 65 (2.0) | |
| Highest level of education | Diploma and lower | 430 (13.3) | 337 (10.5) | 244 7.6) | 1048 (32.5) | 2059 (63.9) | <0.001 |
| | Bachelors and above | 166 (5.2) | 60 (1.9) | 152 (4.7) | 784 (24.3) | 1162 (36.1) | |
| Do you have children, n=3152 | Yes | 51 (1.6) | 54 (1.7) | 108 (3.4) | 300 (9.5) | 513 (16.3) | <0.001 |
| | No | 535 (17.0) | 330 (10.5) | 292 (9.3) | 1488 (47.2) | 2639 (83.7) | |
| Work location | Urban | 339 (10.5) | 49 (1.5) | 299 (9.3) | 550 (17.0) | 1237 (38.3) | <0.001 |
| | Semi-urban | 29 (0.9) | 26 (0.8) | 67 (2.1) | 333 (10.3) | 455 (14.1) | |
| | Rural | 12 (0.4) | 12 (0.4) | 33 (1.0) | 290 (9.0) | 347 (10.8) | |
| | None of the above | 216 (6.7) | 310 (9.6) | 3 (0.1) | 659 (20.4) | 1188 (36.8) | |
| **Total** | | **596 (18.5)** | **397 (12.3)** | **396 (12.3)** | **1832 (56.9)** | **3221 (100.0)** | |

**Table 2. Prevalent estimates of the mental health conditions (depression, anxiety, stress, and any two conditions) and their corresponding Confidence Intervals, by country in sub-Saharan Africa.**

| Mental health | Country | | | | Total |
|---|---|---|---|---|---|
| | Ghana | Malawi | Mozambique | Nigeria | |
| Depression | 40.8 (36.8–44.8) | 72.3 (67.6–76.6) | 59.1 (54.1–63.9) | 52.0 (49.6–54.3) | 53.3 (51.5–55.0) |
| Anxiety | 55.2 (51.1–59.2) | 81.4 (77.2–85.1) | 67.2 (62.3–71.8) | 61.6 (59.3–63.8) | 63.5 (61.8–65.2) |
| Stress | 20.6 (17.5–24.1) | 50.6 (45.6–55.7) | 34.3 (29.7–39.3) | 27.9 (25.9–30.0) | 30.1 (28.6–31.8) |
| Any two | 38.9 (34.9–42.9) | 72.3 (67.6–76.6) | 56.6 (51.5–61.5) | 49.5 (47.1–51.8) | 51.2 (49.5–52.9) |
| All three | 17.3 (14.3–20.6) | 45.3 (40.4–50.4) | 30.6 (26.1 - 35.4) | 25.9 (23.9–28.0) | 27.3 (25.8–28.9) |

For depression, about 23.0% had 'moderate' depression, while 15.5% reported 'mild' depression. In contrast, stress was largely classified as 'normal' (n=2254, 69.9%), with relatively few individuals experiencing severe stress levels.

Fig 2 (bottom panel) shows the percentage of people experiencing mental health disorders by country of origin. Nigeria showed the highest percentage of participants experiencing severe and extremely severe mental health conditions across all categories. For depression and anxiety, a higher percentage of Nigerians reported severe (4.9% and 4.7%) and extremely severe (4.6% and 16.0%) conditions, while Ghanaians reported notably lower proportions, particularly in the severe anxiety category (1.6%). For stress, Nigerian respondents made up the highest percentage of individuals

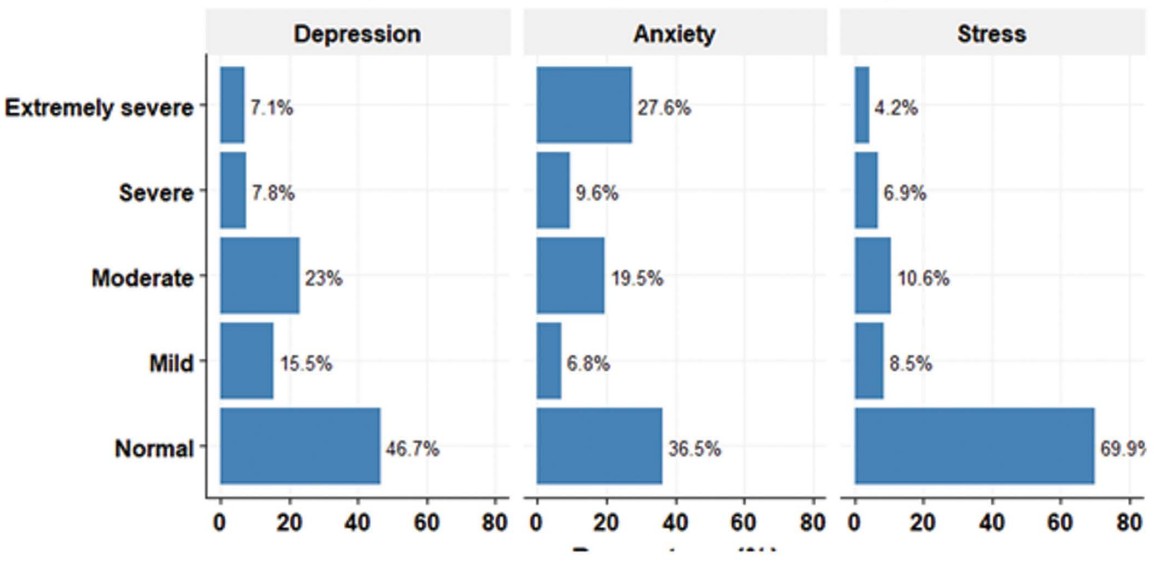

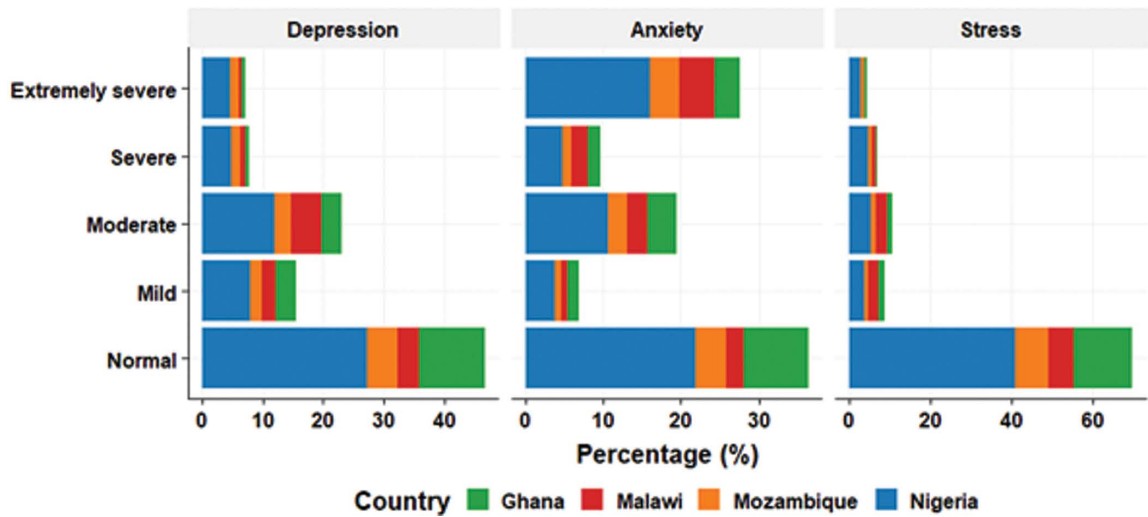

**Fig 2. Overall (top) and country-specific (bottom) severity distribution of depression, anxiety, and stress among staff and students of African universities.**

experiencing severe (4.4%) and extremely severe stress (2.8%), while Ghanaians reported the lowest prevalence of severe (0.6%) and extremely severe stress (0.4%).

### Bivariate Analysis of factors associated with mental health condition by country of origin in SSA

Fig 3 illustrates country-level risk associations with mental health conditions—specifically, depression, anxiety, stress, and combinations of these conditions—with Ghana as the reference category. For depression, individuals in Malawi were 3.79 (95% CI: 2.89 to 5.00) times more likely to experience depression. In comparison, those in Mozambique had odds ratios (OR) of 2.10 (95% CI: 1.62 to 2.72) and in Nigeria, 1.57 (95% CI: 1.30 to 1.90). For anxiety, the odds were 3.54 (95% CI:

## Country-level risk associated with mental health (Univariate)

| Variable | OR | p-value | Odds Ratio | (95% CI) |
|---|---|---|---|---|
| **Depression (Ref: Ghana)** | | | | |
| Malawi | 3.790 | 0.000 | | 3.79 (2.89 to 5.00) |
| Mozambique | 2.098 | 0.000 | | 2.10 (1.62 to 2.72) |
| Nigeria | 1.572 | 0.000 | | 1.57 (1.30 to 1.90) |
| **Anxiety (Ref: Ghana)** | | | | |
| Malawi | 3.542 | 0.000 | | 3.54 (2.64 to 4.80) |
| Mozambique | 1.661 | 0.000 | | 1.66 (1.28 to 2.17) |
| Nigeria | 1.300 | 0.006 | | 1.30 (1.08 to 1.57) |
| **Stress (Ref: Ghana)** | | | | |
| Malawi | 3.944 | 0.000 | | 3.94 (2.99 to 5.23) |
| Mozambique | 2.012 | 0.000 | | 2.01 (1.51 to 2.68) |
| Nigeria | 1.488 | 0.000 | | 1.49 (1.19 to 1.87) |
| **All three (Ref: Ghana)** | | | | |
| Malawi | 3.970 | 0.000 | | 3.97 (2.98 to 5.32) |
| Mozambique | 2.106 | 0.000 | | 2.11 (1.56 to 2.85) |
| Nigeria | 1.675 | 0.000 | | 1.68 (1.33 to 2.13) |
| **Any two (Ref: Ghana)** | | | | |
| Malawi | 4.094 | 0.000 | | 4.09 (3.12 to 5.40) |
| Mozambique | 2.043 | 0.000 | | 2.04 (1.58 to 2.65) |
| Nigeria | 1.535 | 0.000 | | 1.54 (1.27 to 1.85) |

0 1 2 3 4

← Lower risk    Higher risk →

**Fig 3. Country-level risk associations with mental health conditions—specifically, depression, anxiety, stress, and combinations of these conditions—with Ghana as the reference category.** The box size is proportional to the Odd ratio.

2.64 to 4.80) times higher in Malawi, 1.66 (95% CI: 1.28 to 2.17) times higher in Mozambique, and 1.30 (95% CI: 1.08 to 1.57) times higher in Nigeria. For stress, the odds in Malawi were 3.94 (95% CI: 2.99 to 5.23) times higher, in Mozambique 2.01 (95% CI: 1.51 to 2.68) times higher, and in Nigeria 1.49 (95% CI: 1.19 to 1.87) times higher. Regarding the likelihood of experiencing all three conditions simultaneously, Malawi showed the highest odds at 3.97 (95% CI: 2.98 to 5.32), followed by Mozambique at 2.11 (95% CI: 1.56 to 2.85) and Nigeria at 1.68 (95% CI: 1.33 to 2.13). For experiencing any two conditions, the odds were 4.09 in Malawi (95% CI: 3.12 to 5.40), 2.04 in Mozambique (95% CI: 1.58 to 2.65), and 1.54 in Nigeria (95% CI: 1.27 to 1.85).

## Multivariable analysis of factors associated with mental health condition by country of origin in SSA

Table 3 presents the adjusted odds ratios (AORs) for mental health issues across the countries studied after adjusting for socio-demographic variables (See Table 1 for the full variable list). Malawi and Nigeria exhibited the highest and lowest risks respectively of mental health issues compared to Ghana. Specifically, individuals in Malawi were 4.39 times more likely to be depressed, 3.86 (95% CI: 2.81–5.29) more likely to experience anxiety, and 4.51 times more likely to experience stress. They were also 4.67 times more likely to experience all three conditions simultaneously and 4.64 times more likely to face any two conditions.

## Discussion

The study highlights significant variations in mental health outcomes among the higher institutions in the countries studied, with Malawi demonstrating the highest prevalence of mental health conditions. These differences align with existing research [31–33] which attribute disparities in mental health outcomes to factors such as socioeconomic conditions, access to healthcare, social support networks, and cultural attitudes toward mental health [34,35]. The higher prevalence of mental health issues in Malawi may reflect economic hardship, food and job insecurity, and limited healthcare infrastructure [36]. Additionally, cultural perceptions and the stigmas surrounding mental health may influence self-reported symptoms [37] further contributing to the disparities observed between Malawi and other countries, such as Ghana.

Analysis of mental health severity revealed high rates of moderate to severe conditions, especially for anxiety, with 27.7% of respondents experiencing 'extremely severe' anxiety similar to the findings from other studies [38,39]. Although depression was common, it was generally less severe reflecting global trends [40] but also suggests that depressive symptoms are influenced by varying cultural and social factors across countries. The low rate of 'mild' depression may suggest underreporting of early symptoms or poor recognition, with individuals likely seeking treatment only when symptoms worsen [41]. In contrast, most (69.8%) of the respondents had stress levels within 'normal' limits stress levels suggesting that they are relatively more controllable [42,43]. It is likely that, whereas anxiety is more closely associated with feelings of worry and fear, stress may be more dependent on particular situations like social and cultural expectations, economic conditions, and availability of healthcare services [44]. More so, stress may be influenced by coping mechanisms, social support, and workplace conditions, which can differ from one country to another [45]. The severity of mental health conditions was significantly higher in Nigeria, where 16.0% reported extreme anxiety. However, this contrasts with the findings of other studies [46,47] which reported higher levels of extremely severe anxiety and is indicative of country-specific stressors such as economic and political instability, high levels of violence or conflict, or limited access to mental health services [48,49] which may aggravate the mental health conditions.

Sociodemographic factors such as age, gender, and marital status also influenced mental health outcomes in line with existing literature [50,51], with females reporting higher prevalence [52,53]. In this study, students, 2807 (87.0%) and females, 1,850 (57.7%) were the largest subsets of the study population. Studies have shown that conditions such as depression, anxiety, and post-traumatic stress disorder are prevalent among African university students [54], with young women more likely to experience these issues [55,56], due to additional risk factors like poverty, gender-based violence, and societal stigma associated with mental illness [57]. University-based studies in Kenya and South Africa reveal that

Table 3. Adjusted odds ratios and their 95% Confidence intervals (CI) for mental health issues.

| Variables | Depression | Anxiety | Stress | All three | Any two |
|---|---|---|---|---|---|
| Country (ref: Ghana) | AOR (CI) | AOR (CI) | AOR (CI) | AOR (CI) | AOR (CI) |
| Malawi | 4.39 (3.28–5.89) | 3.86 (2.81–5.29) | 4.51 (3.33–6.11) | 4.67 (3.41–6.41) | 4.64 (3.46–6.23) |
| Mozambique | 2.03 (1.55–2.67) | 1.68 (1.27–2.22) | 1.98 (1.46–2.69) | 2.08 (1.51–2.85) | 2.02 (1.54–2.66) |
| Nigeria | 1.59 (1.30–1.95) | 1.30 (1.07–1.59) | 1.48 (1.17–1.88) | 1.70 (1.32–2.18) | 1.53 (1.25–1.87) |

gender-sensitive policies, mental health awareness, and campus support programs can reduce stigma and improve mental health outcomes for female students [58].

In our study, Malawians reported a higher prevalence of mental health disorders compared to other countries even after adjusting for the potential confounders. This could be attributed to the report of an ongoing and severe mental health crisis in Malawi [59] made worse by the poor access to mental health services [60]. Mozambique showed a higher likelihood of depression and anxiety compared to Nigeria, though the impact was less severe than in Malawi. While mental health remains a concern in Mozambique, the lower odds suggest that the crisis may be less intense, possibly due to fewer or more mitigated socioeconomic stressors.

Moreover, comorbidity between depression, anxiety, and stress was common across the sample, with 51.3% of respondents experiencing two of these conditions, and over 25% affected by all three simultaneously. This comorbidity highlights the interconnectedness of these conditions [45], where one disorder can exacerbate the others. Mental health services, therefore, should adopt integrated treatment plans that effectively address multiple conditions simultaneously [61]. This may require collaboration among healthcare providers, including mental health professionals, primary care physicians, and social workers for more comprehensive care that tackles the various aspects of patients' mental health challenges [62]. The considerable overlap among mental health conditions indicates that recognizing and addressing issues at their initial stages could help avert the development of multiple co-occurring disorders.

Country-level analysis revealed distinct patterns in mental health outcomes, with Nigeria showing the highest prevalence of severe mental health disorders. These differences may also reflect varying economic or societal pressures in these countries, with Nigeria potentially experiencing more acute or chronic stressors that contribute to these mental health outcomes [63]. It is acknowledged that the larger sample size of Nigerian participants may contribute to the observed prominence of the country in the severe and extremely severe mental health categories. Conversely, Ghana's relatively favourable mental health outcomes, particularly in anxiety and stress, may reflect protective factors such as stronger social support [64] or the presence of effective coping mechanisms [65]. In addition, it has been noted that while Ghana struggles with under-resourced mental health services, recent policy improvements, such as the 2012 Mental Health Act and ongoing national mental health initiatives, have bolstered its infrastructure slightly [66]. Ghana's relative economic stability and stronger policy frameworks for mental health compared to Malawi may contribute to the differences observed in mental health statistics between the two countries.

Mozambique and Malawi showed elevated rates of two-condition prevalence, particularly in stress and anxiety, suggesting that while these countries face significant mental health challenges, the conditions may not frequently co-occur at extreme levels, as seen in Malawi. Malawi reported the highest rates of depression (72.3%), anxiety (81.4%), and elevated stress (50.6%), surpassing levels observed in other studies [40,67]. In Mozambique, about 21.2% of adolescents experience psychosocial distress due to factors like bullying, violence, and hunger [68], highlighting the urgent need for robust mental health support in schools and communities.

Additionally, cultural views on mental health, including stigma, may discourage individuals from seeking help early, resulting in delayed care until their conditions become severe [69]. Mozambique and Nigeria follow Malawi in the prevalence of two or all three conditions, though at lower rates. Mozambique's notable two-condition prevalence suggests factors that may limit the co-occurrence of all three disorders while Nigeria's higher rates of individual conditions and slightly lower comorbidity imply that mental health disorders there may be more isolated rather than co-occurring.

The findings stress the need for targeted, context-specific mental health interventions in Sub-Saharan Africa. In Malawi, for example, where the mental health burden is severe, resources should prioritize accessible, community-based mental health programs that could leverage existing resources. In Nigeria, anti-stigma campaigns could be developed in collaboration with religious leaders and community organizations to reach a wider audience and address cultural beliefs about mental health and socioeconomic issues such as poverty and unemployment. Mozambique would benefit from strengthened mental health infrastructure and awareness programs, Ghana's comparatively positive mental health outcomes

provide valuable insights that could inform strategies in higher-prevalence countries, particularly around community cohesion and resilience-building. More so, interventions should incorporate mental health literacy at an individual level, to support early detection of mental health issues.

### Strengths and limitations of the study

This study's strengths include a large, diverse sample size and a cross-national focus on Sub-Saharan Africa (SSA). This provides valuable information on mental health across varied socio-economic and cultural backgrounds. The validated DASS-21 scale guaranteed reliable data, and the web-based survey enabled widespread reach. However, the cross-sectional design limits causal inference, and reliance on self-reported data may introduce respondent bias, particularly in populations with strong mental health stigma. While the study's findings provide valuable insights into mental health disparities in SSA, the reliance on convenience sampling limits the generalizability of the results to the wider population. Future studies should consider employing probability sampling techniques to ensure greater representativeness. University students and staff tend to be younger and more educated than the general population, and this may influence their mental health experiences. Incorporating qualitative data from the survey (e.g., open-ended responses) in future studies could provide richer insights into the participants' experiences and perspectives on mental health.

## Conclusions

This cross-sectional study highlights significant cross-national variations in the prevalence of depression, anxiety, and stress among university staff and students in sub-Saharan Africa. The findings emphasize the importance of considering contextual factors, such as education levels, gender, and country of origin, in understanding mental health risks. Many respondents reported experiencing multiple mental health issues simultaneously, suggesting the need for a more holistic treatment approach in the region. Overall, these findings underscore the critical mental health burden in sub-Saharan Africa and the need for increased access to mental health resources and targeted interventions, such as community-based care in Malawi, stigma reduction initiatives in Nigeria, infrastructure development in Mozambique, and resilience-building strategies informed by Ghana's successes.

**Acknowledgments**: The authors acknowledge the support of the centre for Eye Care and Public Health Intervention Initiative (CEPHII) during data collection and thank the following individuals for their contribution during review: Aiyshetu Shuaibu, Godwin O Ovenseri-Ogbomo, Isaac Koomson, and Haile Woretaw.

## Author contributions

**Conceptualization:** Ngozika Esther Ezinne, Edith Daniel-Nwosu, Uchechukwu Levi Osuagwu.

**Data curation:** Oforbuike Onyebuchi Ike, Michael Agyemang Kwarteng, Grace Ogbonna, Isaura Brito dos Santos, Osamudiamen McHillary Ogiemudia, Ellen Konadu Antwi-Adjei, Okechi Ulumma Amaechi, Ejididke Gertrude Ebele, Khathutshelo Percy Mashige, Bernadine Nsa Ekpenyong, Tshubelela Sello Simon Magakwe, Kingsley E. Agho, Ayomikun Nifemi Dahunsi, Oyelola A. Adegboye, Uchechukwu Levi Osuagwu.

**Formal analysis:** Michael Agyemang Kwarteng, Kingsley E. Agho, Oyelola A. Adegboye.

**Investigation:** Isaura Brito dos Santos, Osamudiamen McHillary Ogiemudia, Anayochukwu E. Anyasodor, Ellen Konadu Antwi-Adjei, Okechi Ulumma Amaechi, Ejididke Gertrude Ebele, Ngozika Esther Ezinne, Khathutshelo Percy Mashige, Antor Ndep, Edith Daniel-Nwosu, Bernadine Nsa Ekpenyong, Tshubelela Sello Simon Magakwe, Ayomikun Nifemi Dahunsi, Oyelola A. Adegboye, Kelechi C Ogbuehi, Uchechukwu Levi Osuagwu.

**Methodology:** Oforbuike Onyebuchi Ike, Michael Agyemang Kwarteng, Grace Ogbonna, Osamudiamen McHillary Ogiemudia, Anayochukwu E. Anyasodor, Ellen Konadu Antwi-Adjei, Ejididke Gertrude Ebele, Ngozika Esther Ezinne,

Antor Ndep, Edith Daniel-Nwosu, Bernadine Nsa Ekpenyong, Tshubelela Sello Simon Magakwe, Kingsley E. Agho, Oyelola A. Adegboye, Kelechi C Ogbuehi, Uchechukwu Levi Osuagwu.

**Project administration:** Okechi Ulumma Amaechi, Ngozika Esther Ezinne, Edith Daniel-Nwosu, Uchechukwu Levi Osuagwu.

**Software:** Oforbuike Onyebuchi Ike, Michael Agyemang Kwarteng, Anayochukwu E. Anyasodor, Ngozika Esther Ezinne, Khathutshelo Percy Mashige, Oyelola A. Adegboye, Uchechukwu Levi Osuagwu.

**Supervision:** Grace Ogbonna, Isaura Brito dos Santos, Ellen Konadu Antwi-Adjei, Okechi Ulumma Amaechi, Antor Ndep, Bernadine Nsa Ekpenyong, Kingsley E. Agho.

**Validation:** Osamudiamen McHillary Ogiemudia, Ngozika Esther Ezinne, Edith Daniel-Nwosu, Bernadine Nsa Ekpenyong, Tshubelela Sello Simon Magakwe, Ayomikun Nifemi Dahunsi, Kelechi C Ogbuehi, Uchechukwu Levi Osuagwu.

**Visualization:** Antor Ndep.

**Writing – original draft:** Oforbuike Onyebuchi Ike, Michael Agyemang Kwarteng, Grace Ogbonna, Isaura Brito dos Santos, Osamudiamen McHillary Ogiemudia, Anayochukwu E. Anyasodor, Ellen Konadu Antwi-Adjei, Okechi Ulumma Amaechi, Ejididke Gertrude Ebele, Khathutshelo Percy Mashige, Edith Daniel-Nwosu, Bernadine Nsa Ekpenyong, Tshubelela Sello Simon Magakwe.

**Writing – review & editing:** Oforbuike Onyebuchi Ike, Michael Agyemang Kwarteng, Grace Ogbonna, Ellen Konadu Antwi-Adjei, Okechi Ulumma Amaechi, Ejididke Gertrude Ebele, Ngozika Esther Ezinne, Khathutshelo Percy Mashige, Antor Ndep, Edith Daniel-Nwosu, Bernadine Nsa Ekpenyong, Tshubelela Sello Simon Magakwe, Kingsley E. Agho, Ayomikun Nifemi Dahunsi, Oyelola A. Adegboye, Kelechi C Ogbuehi, Uchechukwu Levi Osuagwu.

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
