## [Decision Letter · Decision Letter 0]

21 Jan 2025

PONE-D-24-56128Cross-National Variations in Mental Health: A Cross-sectional Study on Depression, Anxiety, and Stress in sub-Saharan AfricansPLOS ONE Dear Dr. Osuagwu, Thank you for submitting your manuscript to PLOS ONE. After careful consideration, we feel that it has merit but does not fully meet PLOS ONE’s publication criteria as it currently stands. Therefore, we invite you to submit a revised version of the manuscript that addresses the points raised during the review process.

**ACADEMIC EDITOR: **The manuscript is about cross-national variation in mental health focusing on depression, anxiety and stress among students and staff of tertiary institutions in 4 countries in SSA. This is an important study that will provide evidence toward developing strategies to minimize mental health issues among students and staff. The study however requires some technical adjustment to enable reproducibility and also to strengthen the findings. The authors should please review the following comments as well as the comments from the 2 reviewers below and provide comprehensive response to all comments provided. Methods:

Line 134 – “across several SSA countries”. Comment – if only 4 countries were used, why not indicate “across 4 SSA counties”? Using SSA may be misleading since there is specific number of countries that have participated.Please include “subsection settings” and outline key socioeconomic characteristics of the countries.Inclusion and exclusion criteria – this section is confusing. Please review Lines 182 – 183 versus Lines 186 – 188. The two statements were contradictory.

Results:

Please review plosone guidelines about figures, captions and placement. All figures captions should be labelled in the text where they were firstly mentioned. Submission Guidelines | PLOS ONE

Acknowledgement:

Line 471 – Acknowledgement statement is missing

We look forward to receiving your revised manuscript.

Kind regards,

Ibrahim Jahun, MD, MSC, PhD

Academic Editor

PLOS ONE

Journal Requirements:

2. In the online submission form, you indicated that the data that support the findings of this study are available on request from the corresponding author, ULO.

4. Please ensure the revised manuscript adheres to plosone requirements: https://journals.plos.org/plosone/s/submission-guidelines  

Reviewers' comments:

Reviewer's Responses to Questions

**Comments to the Author**

1. Is the manuscript technically sound, and do the data support the conclusions?

Reviewer #1: Partly

Reviewer #2: Partly

2. Has the statistical analysis been performed appropriately and rigorously? 

Reviewer #1: I Don't Know

Reviewer #2: Yes

3. Have the authors made all data underlying the findings in their manuscript fully available?

Reviewer #1: Yes

Reviewer #2: No

4. Is the manuscript presented in an intelligible fashion and written in standard English?

Reviewer #1: Yes

Reviewer #2: Yes

5. Review Comments to the Author

Reviewer #1: The title should be made to reflection the subpopulation- staff and students, not looking like a community survey.

The introduction did highlight much on literatures that focused on higher institution population as depicted in their objectives.

There is no adequate rationale for this study. Authors should take note.

The study design was merged with study procedure.

The rest of comments are in the pdf comments in the text.

Thanks.

Reviewer #2: 1. Summary of the research and overall impression

The manuscript presents the results of original research into the cross-national variation in mental health: a cross-sectional study on depression, anxiety and stress among sub-Saharan Africans. The authors engaged well with the relevant literature and highlighted the burden of mental health conditions broadly and among students and staff within educational institutions in low-resource settings, justifying the need to generate valid evidence to inform future targeted mental health interventions.

The methods were adequately described, although some conclusions are not supported by the presented results. Overall, the authors engaged well with the topic in the conduct of the research and reporting of the study results. The authors should make a minor revision of the manuscript.

2. Discussion of specific areas for improvement

Having highlighted the strengths and weaknesses of the manuscript above, I have noted here specific areas for improvement by addressing some minor issues:

1. Introduction: The introduction section is thorough, the authors demonstrating a good understanding of the global and local burden of common mental health conditions. However, I have made the following observations:

- Line 122: The statement, “The current body of literature frequently neglects the role of cross-cultural differences among countries” should be supported by a reference.

2. Methods:

- Line 142: The authors noted that the questionnaire was translated and administered in Portuguese and English languages. The authors should have also translated the questionnaire to French, as one of the most common official languages in African countries. Perhaps this could have mitigated the non-response or incomplete responses from some country partcipants.

- Line 166: The authors noted that the DAS-21 was previously validated and pre-tested in their study. It will be helpful to provide the Cronbach’s alpha estimates.

3. Results:

- Figure 2: The caption should be below the figure. The authors should correct this.

- Table 1: The category “Islamic” under religion should be “Islam”

- Line 257: In the statement, “Additionally, 83.7% (n=2639) of respondents indicated that they did not have children”, the proportion should be 81.9% (2639/3221)

- Line 263: Prevalence and distribution of severity of mental health conditions: I suggest the authors present the prevalent estimates of the mental health conditions (depression, anxiety and stress) and their corresponding Confidence Intervals, by country, in a table.

Line 286: “classified as ‘normal’ (N = 2254, 69.9%)” the letter N should be lower case (n) for consistency. The authors should correct this.

- Line 290: The statement, “For depression and anxiety, a higher percentage of Nigerians reported severe (4.9% and 4.7%) and extremely severe (4.6% and 16.0%) depression, while Ghanaians reported notably lower proportions, particularly in the severe anxiety category (1.6%)” should be corrected by replacing “depression” with “conditions”.

- Line 308: For clarity and consistency, I suggest the authors present the Odds Ratios (ORs) with the 95% Confidence Intervals (CIs) together, e.g., 4.09 (95% CI: 3.12 to 5.40) in Malawi, 2.04 (95% CI: 1.58 to 2.65) in Mozambique, and 1.54 (95% CI: 1.27 to 1.85) in Nigeria.

4. Discussion

- Line 430: There should be a full stop after the sentence, “Mozambique would benefit from strengthened mental health infrastructure and awareness programs”.

Conclusions

- Line 450: “The findings from this study suggest a strong link between a person's level of education and their risk for depression, anxiety, and stress”. These conclusions are not supported by the presented results. The authors should review this.

6. PLOS authors have the option to publish the peer review history of their article (what does this mean? ). If published, this will include your full peer review and any attached files.

**Do you want your identity to be public for this peer review?** For information about this choice, including consent withdrawal, please see our Privacy Policy .

Reviewer #1: **Yes: ** Sunday O Oriji

Reviewer #2: **Yes: ** Muftau Mohammed

---

## [Author Response · Author response to Decision Letter 0]

11 Feb 2025

Dear Editor and Reviewers,

Thank you for the time and effort invested in reviewing our manuscript. We appreciate your insightful comments and constructive suggestions, which have helped us improve the quality and clarity of our work. Below, we provide detailed responses to each of the reviewers’ comments. Changes made in the manuscript in response to these comments are highlighted for clarity.

We hope that the revised version of our manuscript meets your expectations and addresses all the concerns raised. The table of responses have been attached in the submitted ‘response to reviewers file’

---

## [Decision Letter · Decision Letter 1]

17 Mar 2025

Cross-National Variations in Mental Health: A Cross-sectional Study on Depression, Anxiety, and Stress among University staff and students in sub-Saharan African

PONE-D-24-56128R1

Dear Dr. Osuagu,

We’re pleased to inform you that your manuscript has been judged scientifically suitable for publication and will be formally accepted for publication once it meets all outstanding technical requirements. Additionally, reviewer 2 raised additional observations which are mainly due to typo and grammar. These observations can be addressed during review process before publication. Kindly endeavour to do so. The observations are:

Title:

- Please change Sub-Saharan African to Sub-Saharan Africa, without an "n"

- Also, please make the "s" in staff, students and sub-Saharan, upper-case letters "S"

Introduction: The issue has been addressed. However, there is an additional observation in Line 119 of the revised version: Please delete the parenthesis before the word, "frequently"

Results:

- Figures 1, 2 and 3 still have captions shown above the figures. These should be corrected.

- Line 276: In the inserted Table 2 title, "prevalent" should be replaced by "prevalence" estimates

Discussion

Line 419: The comma should be replaced by a full stop after “Mozambique would benefit from strengthened mental health infrastructure and awareness programs" before "Ghana's comparatively positive mental health outcomes....

Kind regards,

Ibrahim Jahun, MD, MSC, PhD

Academic Editor

PLOS ONE

Additional Editor Comments (optional):

Reviewers' comments:

Reviewer's Responses to Questions

**Comments to the Author**

1. If the authors have adequately addressed your comments raised in a previous round of review and you feel that this manuscript is now acceptable for publication, you may indicate that here to bypass the “Comments to the Author” section, enter your conflict of interest statement in the “Confidential to Editor” section, and submit your "Accept" recommendation.

Reviewer #1: All comments have been addressed

Reviewer #2: (No Response)

2. Is the manuscript technically sound, and do the data support the conclusions?

Reviewer #1: Yes

Reviewer #2: Yes

3. Has the statistical analysis been performed appropriately and rigorously? 

Reviewer #1: I Don't Know

Reviewer #2: Yes

4. Have the authors made all data underlying the findings in their manuscript fully available?

Reviewer #1: Yes

Reviewer #2: Yes

5. Is the manuscript presented in an intelligible fashion and written in standard English?

Reviewer #1: Yes

Reviewer #2: Yes

6. Review Comments to the Author

Reviewer #1: Dear Editor and Reviewers,

Thank you for the time and effort invested in reviewing our manuscript. We appreciate your insightful comments and constructive suggestions, which have helped us improve the quality and clarity of our work. Below, we provide detailed responses to each of the reviewers’ comments. Changes made in the manuscript in response to these comments are highlighted for clarity.

We hope that the revised version of our manuscript meets your expectations and addresses all the concerns raised.

S/N Line Number Academic Editor’s Comment Response to Reviewer’s Comment

Academic Editors comments

Line 92 – 93 This was not part of your results. How did the authors come to this conclusion?

We have revised the sentence to read: ‘The findings emphasize the importance of considering contextual factors, such as education levels, gender, and country of origin, in understanding mental health risks’.

Line 134 Methods:– “across several SSA countries”. Comment – if only 4 countries were used, why not indicate “across 4 SSA counties”? Using SSA may be misleading since there is specific number of countries that have participated. The study initially involved participants from several SSA countries. However, for the analysis, only data from four countries met the minimum sample size requirement. However, we have revised this section to four SSA countries

Please include “subsection settings” and outline key socioeconomic characteristics of the countries.

A subsection ‘settings’ has been added on the sociodemographic of the countries.

Lines 182 – 183 versus

Lines 186 – 188. Inclusion and exclusion criteria – this section is confusing. Please review The two statements were contradictory. The statement has been revised to read “Survey responses from individuals who were not originally from SSA but were currently studying or working in the region were excluded from the analysis. Additionally, responses originating from identical Internet Protocol addresses and exhibiting similar demographic characteristics, such as age and gender, were considered duplicate entries. In such cases, only the most complete entry was retained for analysis. Participants who provided informed consent were included.”

Results: Please review Plosone guidelines about figures, captions and placement. All figures captions should be labelled in the text where they were firstly mentioned. Submission Guidelines | PLOS ONE

The caption of the figures has been labelled in the text where they were first mentioned.

Reviewer #2: The authors have adequately addressed the issues I raised, but there are a few pending ones, and a new observed one, outlined under the following headings:

Title:

- Please change Sub-Saharan African to Sub-Saharan Africa, without an "n"

- Also, please make the "s" in staff, students and sub-Saharan, upper-case letters "S"

Introduction: The issue has been addressed. However, there is an additional observation in Line 119 of the revised version: Please delete the parenthesis before the word, "frequently"

Results:

- Figures 1, 2 and 3 still have captions shown above the figures. These should be corrected.

- Line 276: In the inserted Table 2 title, "prevalent" should be replaced by "prevalence" estimates

Discussion

Line 419: The comma should be replaced by a full stop after “Mozambique would benefit from strengthened mental health infrastructure and awareness programs" before "Ghana's comparatively positive mental health outcomes.....

7. PLOS authors have the option to publish the peer review history of their article (what does this mean? ). If published, this will include your full peer review and any attached files.

**Do you want your identity to be public for this peer review?** For information about this choice, including consent withdrawal, please see our Privacy Policy .

Reviewer #1: **Yes: ** DR S O ORIJI

Reviewer #2: **Yes: ** Muftau Mohammed

---

## [Editor Report · Acceptance letter]

PONE-D-24-56128R1

PLOS ONE

Dear Dr. Osuagwu,

I'm pleased to inform you that your manuscript has been deemed suitable for publication in PLOS ONE. Congratulations! Your manuscript is now being handed over to our production team.

Kind regards,

on behalf of

Dr. Ibrahim Jahun

Academic Editor

PLOS ONE